# Potential for the Production of Carotenoids of Interest in the Polar Diatom *Fragilariopsis cylindrus*

**DOI:** 10.3390/md20080491

**Published:** 2022-07-29

**Authors:** Sébastien Guérin, Laura Raguénès, Dany Croteau, Marcel Babin, Johann Lavaud

**Affiliations:** 1IRL7266 Takuvik, CNRS (France)/ULaval (Canada), Pavillon Alexandre-Vachon, Université Laval, 1045, av. de la Médecine, Québec, QC G1V 0A6, Canada; ragueneslaura@gmail.com (L.R.); dany.croteau@ibpc.fr (D.C.); marcel.babin@takuvik.ulaval.ca (M.B.); johann.lavaud@bio.ulaval.ca (J.L.); 2UMR6539 LEMAR-Laboratory of Environmental Marine Sciences, Institut Universitaire Européen de la Mer, CNRS, IRD, Ifremer, Université de Brest, 29280 Plouzané, France

**Keywords:** polar diatoms, fucoxanthin, diadinoxanthin, diatoxanthin, blue light, photosynthesis

## Abstract

Carotenoid xanthophyll pigments are receiving growing interest in various industrial fields due to their broad and diverse bioactive and health beneficial properties. Fucoxanthin (Fx) and the inter-convertible couple diadinoxanthin–diatoxanthin (Ddx+Dtx) are acknowledged as some of the most promising xanthophylls; they are mainly synthesized by diatoms (Bacillariophyta). While temperate strains of diatoms have been widely investigated, recent years showed a growing interest in using polar strains, which are better adapted to the natural growth conditions of Nordic countries. The aim of the present study was to explore the potential of the polar diatom *Fragilariopsis cylindrus* in producing Fx and Ddx+Dtx by means of the manipulation of the growth light climate (daylength, light intensity and spectrum) and temperature. We further compared its best capacity to the strongest xanthophyll production levels reported for temperate counterparts grown under comparable conditions. In our hands, the best growing conditions for *F. cylindrus* were a semi-continuous growth at 7 °C and under a 12 h light:12 h dark photoperiod of monochromatic blue light (445 nm) at a PUR of 11.7 μmol photons m^−2^ s^−1^. This allowed the highest Fx productivity of 43.80 µg L^−1^ day^−1^ and the highest Fx yield of 7.53 µg Wh^−1^, more than two times higher than under ‘white’ light. For Ddx+Dtx, the highest productivity (4.55 µg L^−1^ day^−1^) was reached under the same conditions of ‘white light’ and at 0 °C. Our results show that *F. cylindrus*, and potentially other polar diatom strains, are very well suited for Fx and Ddx+Dtx production under conditions of low temperature and light intensity, reaching similar productivity levels as model temperate counterparts such as *Phaeodactylum tricornutum*. The present work supports the possibility of using polar diatoms as an efficient cold and low light-adapted bioresource for xanthophyll pigments, especially usable in Nordic countries.

## 1. Introduction

Carotenoid xanthophyll pigments are receiving growing interest in various industrial fields such as the functional foods industry, nutraceuticals and cosmetics, etc. [1,2,3,4], with an exponentially growing global market value estimated to be worth >100 millions of dollars [2,3]. Rapidly emerging interest in xanthophylls is due to their broad and diverse bioactive and health beneficial properties, including antioxidant, anti-inflammatory and anti-obesity [1,2,3,5]. Similarly of major interest is their anti-proliferative and anticancer activity against various cancer cell lines [1,4,6]. For these reasons, fundamental and applied research on xanthophyll pigments has increased exponentially in recent years [7].

Fucoxanthin (Fx, Figure 1) and the inter-convertible couple diadinoxanthin–diatoxanthin (Ddx+Dtx, Figure 1) are acknowledged as some of the most promising xanthophylls [3,4,6,8]. Fucoxanthin and Ddx+Dtx are mainly synthesized by representants of the stramenopile clade, within which diatoms (Bacillariophyta) comprise the most productive group [9]. Thanks to diatoms’ worldwide abundance and crucial role in the marine primary production of diatoms [10], Fx is regarded as the most abundant pigment in the world’s ocean after chlorophylls [11]. Fx is a major ‘accessory’ light-harvesting pigment, while Ddx+Dtx play a double-faceted role in light harvesting and the dissipation of excess light energy (‘photoprotection’) depending on growth conditions [12], i.e., typically, Ddx+Dtx are synthesized under stressful conditions and/or when cell growth rate is slowed down.

Originally, Fx was commercially produced from brown seaweed [1,13,14]; nowadays, and in order to meet the global demand, the use of diatoms is increasing. Aside from a 10 times higher Fx content (mg g^−1^) [3], the main advantages of using diatoms is their higher growth rate, their ability to grow both outdoors and indoors under controlled conditions, and the ease with which they can be genetically manipulated, in particular for the genetic engineering of xanthophyll production [15,16,17,18]. While temperate strains of diatoms have been widely investigated, especially those easily growing model strains with a sequenced genome such as *Phaeodactylum tricornutum* [19], recent years showed a growing interest in using polar strains, which are better adapted to the growth conditions of Nordic countries [20,21,22,23]. Additionally, the adaptation of polar strains to their extreme environment, in particular to low temperatures and irradiances, has been accompanied by their cell enrichment in specific molecules and metabolites such as the xanthophylls and polyunsaturated fatty acids [24], conferring a unique biochemical diversity and a broad bioactivity [25,26]. This is especially the case for Fx, which is sometimes found to be as abundant as chlorophyll *a* [27]. It is foreseen that these characteristics could satisfy, for instance, the outdoor environmental constraints for the production of valuable algal biomass in northern regions [20,21,22,28,29,30], as well as during winter at more temperate latitudes. Therefore, polar diatoms are being studied as an important cold and low light-adapted biotechnological resource, especially usable in Nordic countries, to develop new economics based on marine resources [21,23].

In addition to pioneer works in genetic engineering [3,15] and synthetic biology [31], the improvement of xanthophyll synthesis can also be achieved by screening for naturally highly producing diatom strains [29,32,33,34,35,36] and/or of specific growth conditions, including the light climate (photoperiod, intensity, spectrum), nutrient availability, temperature, and salinity, (see the recent syntheses [3,4,13,14]. As regards to light, a trade-off between cell growth and Fx and Ddx+Dtx synthesis has been reported, and is different from one species to another (see [37]). Additionally, blue wavelengths, the spectral region that is best absorbed by xanthophylls, tend to stimulate Fx and Ddx+Dtx synthesis to an extent that is intensity- and species-dependent [35,37,38,39,40,41,42]. The adaptation of polar diatoms to a low-light environment enriched in blue-green wavelengths (typically in and underneath sea-ice [43,44]) thus suggests that the use of blue light of low intensity may boost their xanthophyll synthesis while keeping cell growth maximal. Finally, temperature in the range 0–10 °C has been shown to differentially impact on the Fx and Ddx+Dtx cell contents [45,46].

The aim of the present study was to explore the potential of the polar diatom *Fragilariopsis cylindrus* in producing Fx and Ddx+Dtx. *F. cylindrus* is a ubiquitous cold water-adapted model species. Its genome has been sequenced [47], its metabolic network modeled [48], and its photobiology and response to environmental conditions (e.g., nutrients, trace metals, temperature, salinity) partially described [45,49,50,51,52].

In order to find the best xanthophyll producing conditions for *F. cylindrus*, it was grown under various light conditions (length of photoperiod, and light intensity and spectrum) and temperatures. We then compared its optimal xanthophyll production rate to the largest ones reported for temperate counterparts grown under autotrophic conditions [35]. 

## 2. Results

### 2.1. Production of Fucoxanthin (Fx) and Diadinoxanthin+Diatoxanthin (Ddx+Dtx) in Fragilariopsis cylindrus under a Range of Photoperiods

In a first step, we investigated the Fx and Ddx+Dtx cell contents and productivity of *Fragilariopsis cylindrus* grown in semi-continuous mode at 0 °C across a range of photoperiods with the same light intensity (30 µmol photons m^−2^ s^−1^ of photosynthetically available radiation (PAR), i.e., the optimal irradiance for *F. cylindrus* growth [52,53] and ‘white’ light spectrum (which reproduces the bottom ice light spectrum at best, see Appendix A) (Figure 2). While the growth rate increased as a function of photoperiod length up to around 0.25 day^−1^ for a 18 h daylength and thereafter was saturated, F_V_/F_M_ remained between 0.62 and 0.66 regardless of the photoperiod (Figure 2a). We observed a maximized Fx content per dry weight (DW) and productivity (µg L^−1^ day^−1^) with a 12 h light:12 h darkness (12 L:12 D) photoperiod, i.e., content and productivity did not increase with further increasing the daylength (Figure 2b). 

The Ddx+Dtx content and productivity were the highest under a 24 L:0 D photoperiod and decreased with the daylength (except in complete darkness (0 L:24 D)) (Figure 2c). When grown in batch mode under the same light climate, we confirmed that the Fx and Ddx+Dtx contents of cultures (i.e., mg L^−1^), as well as the photosynthetic performance (F_V_/F_M_) of *F. cylindrus*, were higher under a 12 L:12 D vs. an 18 L:6 D photoperiod (Appendix A). Under these conditions, we also observed a saturation of the Fx accumulation for cell concentrations over 1.5 × 10^6^ cells mL^−1^. Therefore, the following experiments were performed with a 12 L:12 D photoperiod at 0 °C and in semi-continuous culturing mode to maintain a cell density of around 1 × 10^6^ cells mL^−1^ and stable optical properties of cultures, except when indicated otherwise. 

### 2.2. Production of Fx in F. cylindrus under Different Light Spectra

The second set of experiments consisted of varying the light spectrum with a constant photosynthetically usable radiation (PUR) of 11.7 µmol photons m^−2^ s^−1^ (corresponding to a PAR of 30 µmol photons m^−2^ s^−1^ of ‘white’ light (see Table 1). Three light spectra were compared: ‘white’, blue (peaking at 445 nm) and red (peaking at 660 nm) (Figure 3, see Table 2 for the definition and units of all parameters measured and Appendix A for their values). 

Under these conditions, the growth rate, cellular Chl *a* and C contents, F_V_/F_M_ and all photosynthetic parameters (especially Y_PSII_, Appendix A) of red light-grown cells were lower. This was paralleled by a higher non-photochemical quenching (NPQ_gE_ and NPQ_max_) (Figure 3a, Appendix A) and non regulated non-photochemical enegry losses (Y_NO_, Appendix A). The highest growth rate was found in blue light-grown cells (0.26 ± 0.04 day^−1^), with no significant difference in F_V_/F_M_ and NPQ_gE_ compared to ‘white’ light-grown cells (Figure 3a). While the Fx content per unit of dry weight of microalgae was significantly higher under red light (8.95 ± 1.77 mg g^−1^), there was no difference between blue and ‘white’ light (Figure 3b). Nevertheless, due to the very low growth rate, both the Fx productivity and production yield were dramatically lower under red light. Conversely, they were the highest under blue light, at 21.42 ± 2.56 µg L^−1^ day^−1^ and 3.71 ± 0.44 µg Wh^−1^, respectively, being about two (Fx productivity) and three (Fx yield) times higher than under ‘white’ light.

### 2.3. Production of Fx in F. cylindrus under Different Light Intensities

The third set of experiments consisted of lowering the light intensity from a PUR of 11.7 to 5.8 µmol photons m^−2^ s^−1^ (corresponding, respectively, to a PAR of 30 and 15 µmol photons m^−2^ s^−1^ of ‘white’ light, see Table 1) under ‘white’ and blue (445 nm) spectra (Figure 4, Appendix A). As expected, under 5.8 µmol photons m^−2^ s^−1^, the growth rate was significantly lower, as well as rETR_max_ and E_k_, and Chl *a*/C doubled (Appendix A). The highest Fx content (7.80 ± 0.50 mg g^−1^) and productivity (21.42 ± 2.56 mg g^−1^) were found in cells growing at a PUR of 5.8 µmol photons m^−2^ s^−1^ and 11.7 µmol photons m^−2^ s^−1^ of blue light, respectively (Figure 4b). The Fx yield of blue light-grown cells was three times higher than that of ‘white’ light growing cells, independent of the PUR intensity (Figure 4b).

### 2.4. Production of Fx in F. cylindrus under Blue Light of Different Doses

In the fourth set of experiments (Figure 5, Appendix A), using blue light only, the light dose (cumulative light intensity over a day) was increased by doubling the PUR to 23.4 µmol photons m^−2^ s^−1^ or by doubling the daylength to a 24 L:0 D photoperiod while maintaining a PUR of 11.7 µmol photons m^−2^ s^−1^ (see Table 1). The growth rate, F_V_/F_M_ (Figure 5a ), and photosynthetic parameters (especially rETR_max_, Appendix A) of cells growing under 24 L:0 D were significantly lower compared to a 12 L:12 D photoperiod for the same PUR (11.7 µmol photons m^−2^ s^−1^), with no difference in Fx content, but lower productivity and yield. When doubling the PUR for a 12 L:12 D photoperiod, the growth rate did not significantly differ from that at lower PURs (11.7 and 5.8 µmol photons m^−2^ s^−1^). Under the highest tested PUR of 23.4 µmol photons m^−2^ s^−1^, the Fx content, productivity and yield were lower (Figure 5), the cellular Chl *a* and C contents as well as F_V_/F_M_ and all photosynthetic parameters significantly decreased, while all NPQ parameters (NPQ_gE_, NPQ_max_, Y_NPQ_) increased (Appendix A).

### 2.5. Production of Fx in F. cylindrus under Different Temperatures

The fifth set of experiments consisted of increasing the growing temperature (from 0 °C to 7 °C) with a PUR of 11.7 µmol photons m^−2^ s^−1^ under ‘white’ and blue light spectra (Figure 6). At 7 °C, the cells grew significantly faster under blue light; they showed no significant differences compared with the other growth conditions (0 °C and 7 °C ‘white’ light) (Figure 6a), except for two noticeable differences: blue light-grown cells showed the highest Y_PSII_ (0.49 ± 0.02) and the lowest Y_NPQ_ (0.03 ± 0.01) among all growing conditions (Appendix A). The 7 °C ‘white’ light-grown cells showed significantly lower Chl *a* and C cell contents, F_V_/F_M_, and ETR parameters than at 0 °C, while the NPQ and PSII energy partitioning parameters were similar (Figure 6a, Appendix A). While there were no significant differences in Fx content at 7 °C, the Fx productivity and yield doubled for the 7 °C blue light-grown cells compared to 0 °C.

### 2.6. Production of Ddx+Dtx in F. cylindrus

In parallel with Fx, the same parameters were computed for Ddx+Dtx (Figure 7). Here, the highest Ddx+Dtx content (1.73 ± 0.52 mg g^−1^) but the second lowest Ddx+Dtx productivity (0.79 ± 0.62 µg L^−1^ day^−1^) and lowest production yield (0.09 ± 0.07 µg Wh^−1^) were recorded for 0 °C red light-grown cells. The other conditions for high Ddx+Dtx content were 0 °C and blue light with PUR values of 11.7 and 23.4 µmol photons m^−2^ s^−1^. The highest values of Ddx+Dtx productivity (4.55 ± 0.36 µg L^−1^ day^−1^) were found under ‘white’ light with a PUR of 11.7 µmol photons m^−2^ s^−1^ at 0 °C. The significantly highest Ddx+Dtx production yields were observed for 0 °C ‘white’ light with a PUR of 5.8 and 11.7 µmol photons m^−2^ s^−1^, and for blue light with a PUR of 11.7 µmol photons m^−2^ s^−1^ at 0 °C and 7 °C.

### 2.7. Synthesis of All Conditions Tested for the Production of Fx and Ddx+Dtx in F. cylindrus

When pooling all conditions and parameters together (Appendix A), the cells grown at 7 °C showed the highest growth rate, Fx productivity and Fx yield, whereas the cells grown at 0 °C showed the highest Ddx+Dtx content and NPQ_gE_ (Appendix A). The growth rate, Fx productivity and yield were maximized under blue light. On the other hand, under red light, the Fx and Ddx+Dtx contents and NPQ_gE_ were maximal, while this condition promoted the lowest growth rates, F_V_/F_M_, and Ddx+Dtx productivity and yield (Appendix A). While there were no significant differences in Fx and Ddx+Dtx productivity between light intensities, the strongest Fx and Ddx+Dtx yields were found in cells growing at a PUR of 11.7 µmol photons m^−2^ s^−1^ (equivalent to PAR 30 µmol photons m^−2^ s^−1^) (Appendix A).

### 2.8. Fx producion in F. cylindrus Compared with Temperate Counterparts

The last set of experiments consisted of growing *F. cylindrus* in similar conditions as diatom temperate counterparts as in [35], but under the lower temperature and light intensity our polar strain is adapted to (i.e., 0 °C/7 °C instead of 20 °C, and a PAR of 30 instead of 80 µmol photons m^−2^ s^−1^). Growing conditions were therefore: batch culturing, ‘white’ light, PUR of 11.7 µmol photons m^−2^ s^−1^ and varying both the temperature and the culturing medium as follows: 0 °C + f/2, 0 °C + f, 7 °C + f. The growth rate was significantly higher in cells growing in f medium at 0 °C and 7 °C than in f/2 medium at 0 °C (Figure 8a,b). Although Fx and Ddx+Dtx (Figure 8c,d) contents were not significantly different between the three growing conditions, Fx and Ddx+Dtx productivity was maximized in f medium 7 °C-grown cells (Figure 8).

## 3. Discussion

### 3.1. Acclimation of Fragilariopsis cylindrus to Different ‘White’ and Blue Light Photoperiods, and Intensities, and Effect of the Temperature

When culturing *F. cylindrus* at 0 °C under increasing daylength of ‘white’ light at a PUR of 11.7 µmol photons m^−2^ s^−1^, we observed a proportional growth rate increase with daylength until it reached a maximum with 18 h of light. The light intensity being fixed, longer daylength results in stronger daily light dose, which even under moderate light intensity can result in higher pressure on PSII at low temperature [54], leading to an increased risk of photodamage and photoinactivation [55]. The higher light dose’s effect on PSII is exacerbated by the shortening of the darkness period by reducing cells’ ability to repair damaged PSII reaction centre D1 protein [56], leading to a lowering of active PSII and photosynthesis efficiency, and higher metabolic costs [54,55,56,57], ultimately illustrated by the saturation of the growth rate. One of the methods diatoms use to limit this situation is to dissipate the excess light energy via the non-photochemical quenching (NPQ) [58,59], as shown by the increase in NPQ_gE_ and its regulatory partner Ddx+Dtx with daylength, reaching a maximum under 24 h of light, similarly to as reported before [27,51].

In our study, a PUR of 11.7 µmol photons m^−2^ s^−1^ translates into a PAR of 30 µmol photons m^−2^ s^−1^ when ‘white’ light is used, which matches the light optimum of *F. cylindrus* [52,53]. Lowering the PUR to 5.4 µmol photons m^−2^ s^−1^ under both ‘white’ and blue lights for the same photoperiod (12 h L:12 h D) led to a lower growth rate accompanied by a decrease in photosynthesis (rETR_max_, α, E_k_, E_opt_), an increase in light harvesting pigments (Chl *a* and Fx cell contents), and no change in the dark-acclimated photochemical efficiency (F_V_/F_M_) and NPQ_gE_, which is a typical photoacclimatory response to a non-saturating light intensity in polar diatoms [27,53,60,61,62].

The higher light conditions we tested with the same photoperiod (23.4 µmol photons m^−2^ s^−1^ PUR) exceeded the optimal growth light intensity of *F. cylindrus* at 0 °C [53]. In order to optimise the light energy usage and to maintain the same growth rate, *F. cylindrus* responded by lowering its light harvesting capacity (decrease in the Chl *a* and Fx contents), and photochemical efficency under light limitation (α), and increasing its capacity for dissipating the excess light energy (doubling of NPQ_gE_, increase in NPQ_max,_ and Y_NPQ_), which partially resulted in the decrease in their photosynthetic efficiency (F_V_/F_M_, rETR_max_). This is typical of high-light response in polar diatoms [50,60].

Interestingly, when comparing blue and ‘white’ light treatments of the same PUR (5.8 and 11.7 µmol photons m^−2^ s^−1^ PUR), we did not observe (apart from a consistent but not significant increase in NPQ) the blue light-triggered high-light acclimation previously described in temperate diatoms [38,63] and shown to be controlled by the aureochrome blue light photoreceptor family [64,65], although *F. cylindrus* does possess aureochromes [66]. *F. cylindrus* is usually found to be abundant at the sea-ice bottom horizon and underneath ice [51,53] where blue wavelengths dominate [43,44]. We propose that the apparent absence of this specific blue light response could be due to a differential sensitivity to blue light as compared to the planktonic temperate diatom species examined before [38,63]. This possibility is supported by the fact that increasing the daylength from 12 h to 24 h (PUR 11.7 µmol photons m^−2^ s^−1^) did not trigger any major change in NPQ (NPQ_gE_, NPQ_max_, Y_NPQ_). 

The positive effect of the temperature increase from 0 °C to 7 °C on growth and photosynthesis was similar that one reported before [49,67,68]. At 7 °C, the growth rate increased especially under blue light, with a major increase in photosynthetic efficiency (Y_PSII_, α), while the cells’ Chl *a* content and photoprotective capacity decreased (i.e., lower NPQ_max_ and halved Ddx+Dtx). This phenotype can be accompanied by a typical high-to-low light intensity response [46,69]. Both decreasing the light intensity and increasing temperature can leverage excitation pressure on PSII either directly (low light) or indirectly, i.e., a higher temperature can alleviate downstream bottlenecks by increasing carbone fixating enzymes rates. The 7 °C growth rate was nearly doubled under blue compared to ‘white’ light, with no differences in photosynthesis and photoprotection, a feature also observed at 0 °C (11.7 µmol photons m^−2^ s^−1^ PUR) but to a lower extent. This observation was also reported in temperate counterparts, and shown to be based on the cell cycle and sexual reproduction [66,70] and/or on improved energy vs. biomass allocation [38,39,63].

### 3.2. The Unique Response of F. cylindrus to Red Light

When using equivalent PURs between light treatments, a slower growth rate, due to a higher quantum requirement, is to be expected under monochromatic red light in diatoms [38,64,71,72]. The dramatically slower growth rate (close to 0 day^−1^) was similar to previous reports in temperate planktonic and benthic strains [66]. Nevertheless, apart from this observation and similar cell Chl *a* content with ‘white’ and blue lights, the response of *F. cylindrus* to red light appeared stronger than in temperate strains [38,64]. It showed a dramatic drop in light usage, photosynthetic efficiency (lower F_V_/F_M_, rETR_max_, α, Y_PSII_) and cell C content paralleled with a larger pool of xanthophylls Ddx+Dtx and a doubled NPQ_gE_ (but a similar NPQ potential, NPQ_max_). In fact, *F. cylindrus*’ response to low PUR 12:12 red light resembled that of a 24:0 light cycle or a high intensity of red light in *P. tricornutum* [72], which is linked to a lower potential for PSII photodamaged repair and for reactive O_2_ species (ROS) scavenging (high Y_NO_), although xanthophyll synthesis and NPQ capacity are maintained [72].

In temperate strains, red light is mainly sensed by phytochrome photoreceptors [73], and low fluences were proposed to help sensing the red wavelength-enriched surface of mixed water columns, and potentially higher light exposure [73], and were shown to trigger the photoprotective process (i.e., induction of the Ddx de-epoxidation and NPQ of ‘white’ light-acclimated cells [38]). While such roles can be questioned in *F. cylindrus* where no homologous phytochromes genes have been found so far [74], the long-term (several weeks) photoacclimatory response we observed points to a similar short-termed (4 days at max) higher red light dose in *P. tricornutum* [72]. We propose that this seemingly exacerbated response may be due to three non-exclusive facts. Fact 1: the 0 °C temperature at which the experiments were performed, i.e., cold adaptation in polar diatoms was proposed to rely on similar physiological processes as high light acclimation [49] as in other photosynthetic organisms [75]. Fact 2: *F. cylindrus* is usually found to be abundant at the sea-ice bottom horizon and underneath ice [51,53], where red wavelengths are largely attenuated [43,44], and thus a 12 h monochromatic red light exposure per day during several weeks might have triggered the dramatic response we report. Fact 3: red light does not occur alone without a significant amount of blue-green light in the natural habitat of *F. cylindrus*, and the lack of blue light, especially, could be partly responsible for its poor performance under monochromatic red light. 

### 3.3. Fucoxanthin Production in F. cylindrus

In a first attempt, we grew *F. cylindrus* in batch mode at 11.7 µmol photons m^−2^ s^−1^ PUR of ‘white’ light. As reported before for other species [32,33,76], these batch cultures showed their highest volumetric Fx concentration and F_V_/F_M_ by the end of the exponential phase, before declining with high cell density and nutrient depletion. Fx productivity is therefore closely related to diatom growth [77], and keeping the cells in exponential growth phase is crucial to maximizing Fx production as well as maintaining their photosynthetic capacity. This first experiment provided the optimal cell concentration range (7.5 × 10^5^ to 1.0 × 10^6^ cells mL^−1^) to maintain exponential phase and maximal Fx productivity in semi-continuous growing conditions. Additionally, as described above (see Section 3.1), the response of *F. cylindrus* to a range of photoperiods provides two important conclusions: (i) *F. cylindrus* grows efficiently under a broad range of daylengths at 0 °C, making it suitable for biomass production under outdoor solar light at high latitudes where seasonal daylength varies widely and rapidly, and (ii) a daylength longer than 12 h is not beneficial in terms of energy costs vs. Fx productivity. 

Under these growing conditions (semi-continuous, 12 h light:12 h dark photoperiod, ‘white’ light), the Fx contents were similar to the ones previously reported for *F. cylindrus* at similar light intensities (30 µmol photons m^−2^ s^−1^ PAR) and temperatures (between 0 °C and 7 °C) [45,49,53,60]. Because several studies also previously showed, in temperate diatoms, that blue light increases carotenoid and Fx production [35,38,78], we tested that option. In our conditions (445 nm from 5.8 to 23.4 µmol photons m^−2^ s^−1^ PUR), *F. cylindrus* showed 2-fold higher Fx production as compared to ‘white’ light growing cells. The blue light positive effect on Fx synthesis was consistent regardless of the growing light intensity and temperature. Furthermore, the very high absorption coefficient of *F. cylindrus* around 445 nm (90%, Appendix A) resulted in lowering blue LED intensity (13 instead of 30 µmol photons m^−2^ s^−1^ PAR) for reaching the same productivity as under ‘white’ light. For that matter, coupling a halved light energy consumption with a higher productivity generated a Fx yield consistently three times higher in the blue light growing cells. Contrary to several previous reports [39,76,79], lowering the light intensity did not trigger the expected increase in Fx, but rather the opposite. This is likely due to the fact that the lowest intensity we tested (5.8 μmol photons m^−2^ s^−1^ PUR) is much lower than those used before (see Appendix A) and it strongly limits cell growth and Fx production, even in the low light-adapted *F. cylindrus* [45,51,53]. Conversely, and as expected, increasing the light intensity (to 23.4 μmol photons m^−2^ s^−1^ PUR) decreased Fx production to a similarly low level as with low light. Finally, increasing *F. cylindrus* growth temperature from 0 °C, which is the closest temperature to its natural habitat, to 7 °C, resulted in the highest growth rate, Fx productivity and yield, likely thanks to an accelerated metabolism [21,45,46,49]. Hence, in our hands, the best growing conditions for *F. cylindrus* were a semi-continuous growth at 7 °C and under a 12 h light:12 h dark photoperiod of monochromatic blue light (445 nm) at a PUR of 11.7 μmol photons m^−2^ s^−1^. This allowed the highest Fx productivity of 43.80 µg L^−1^ d^−1^. However, we propose that a stronger light dose (by increasing light intensity or daylength) would probably drive an even higher Fx productivity at 7 °C [46].

### 3.4. Diadinoxanthin–Diatoxanthin Production in F. cylindrus

In contrast to Fx (see Appendix A), and although they show promising bioactive properties [6], to the best of our knowledge, productivity data on the xanthophyll Ddx+Dtx are unavailable. All available data indeed refer to cellular content or content relative to Chl *a* (see the recent synthesis by [12]). Fx and Ddx+Dtx do not have the same function: Fx is a light-harvesting pigment while Ddx+Dtx mainly serve as a strong antioxidant to scavenge ROS and as a modulator of the NPQ process [12,58]. There is a light dose and growth rate trade-off between the synthesis of Ddx+Dtx and of Fx, which reflects the fine balance between light harvesting, photosynthetic electron transport, C-metabolism and growth (see for instance [80,81]). When electron transport exceeds C-fixation capacities, it can be downregulated by an increase in NPQ (positively correlated with the Ddx+Dtx content) and by a decrease in light absorption through a decrease in Chl *a* and Fx contents. Usually, cells growing under optimal (i.e., close to E_k_) light levels do not show a high Ddx+Dtx content and neither induce NPQ. While this is generally true for planktonic strains, this is less the case for benthic forms [82] and for polar strains [12,27,51,53], likely because of low light and/or low temperature adaptation [61]. Therefore, the growing and light conditions for the optimal production of Fx and Ddx+Dtx are not necessarily similar [12,33,37,83], and the growing conditions for reaching the highest Ddx+Dtx cell content or the strongest productivity can be different [12]. For instance, high Ddx+Dtx cell content was reported when (i) slowing down the growth rate via drastically decreasing the light dose with intermittent light (i.e., 5 min light per hour; [80,81]); (ii) manipulating the red compound of the light spectrum or the blue:red light ratio [37,38,40,72,84]; and (iii) exposing the cells to a stress, such as an excess of light [27,53,82]. In the present study, we observed this diversity of response which is promising for future manipulation of Ddx+Dtx cell content and/or productivity alone or along with Fx (see above Section 3.3). The highest Ddx+Dtx biomass content was obtained under red light, most likely due to the close to 0 day^−1^ cell division, therefore yielding the lowest Ddx+Dtx productivity, similar to (as expected) that obtained for the lowest PUR of blue light (5.8 µmol photons m^−2^ s^−1^). Contrastingly, the highest Ddx+Dtx productivity was obtained with ‘white’ light growing cells at 0 °C with optimal light conditions (i.e., 12 h light:12 h dark photoperiod and 11.7 µmol photons m^−2^ s^−1^ PUR). Monochromatic blue light was not as efficient in producing Ddx+Dtx than Fx, pointing to the importance of the blue:red ratio [38,40]. Nevertheless, blue light Ddx+Dtx productivity could be enhanced by increasing PUR (to 23.4 µmol photons m^−2^ s^−1^), while increasing the daylength (to 24 h) or temperature (to 7 °C) had no (daylength with blue light) or a deleterious (temperature with ‘white’ light) effect. Interestingly, under blue light, the temperature increase yielded the same Ddx+Dtx productivity as under 0 °C and multiplied that of Fx by two. In the context of Ddx+Dtx and Fx co-production, we furthermore observed that (i) longer ‘white’ light daylength increases Ddx+Dtx production together with a stable Fx production (Figure 2), and (ii) Ddx+Dtx volumetric accumulation in *F. cylindrus* batch cultures can be increased alongside with maintaining the one of Fx by waiting for the cultures to reach their stationary phase of growth (Appendix A), a feature also reported before for temperate diatoms [85].

### 3.5. Comparison of F. cylindrus Fucoxanthin Production with Temperate Counterparts: Maximization through Growing Conditions

As expected, *F. cylindrus* is effective in biomass and Fx production under light and temperature conditions that are close to its natural growing conditions: low temperature (0 °C) and low light (PAR 30 μmol photons m^−2^ s^−1^ and 13 μmol photons m^−2^ s^−1^ of ‘white’ and blue light, respectively) [61]. Yet, *F. cylindrus* showed slower growth rates, biomass and Fx production as compared to those generally reported in temperate strains (see Appendix A). The slower growth rate is characteristic of most polar diatoms [53], as compared to many planktonic and some small benthic forms of temperate diatoms [61]. Since higher biomasses, similar to those obtained with temperate diatoms, can lead to higher Fx production, we grew *F. cylindrus* in batches under different temperature and nutritive conditions similar to the ones used for temperate counterparts in [35]. *F. cylindrus* biomass accumulation was similar to planktonic *Phaeodactylum tricornutum* and *Thalassiosira weissflogii* [35,76,86]. The Fx content was higher compared to semi-continuous growth (see above Section 3.3), and was similar to *P. tricornutum* and *T. weissflogii*, and benthic *Amphora sp* [35,87], and superior to most diatoms in a non enriched culture medium (see Appendix A and Wang et al. [88]). Nevertheless, *F. cylindrus* Fx productivity remained three times lower than these three species, 10 times lower than the strong producer *Cylindrotheca closterium* [35,89], and 70 times lower than the most Fx productive *Ondotella aurita* (Appendix A) [32,78].

However, polar diatoms remain relevant to producing Fx, and other high-added value compounds [29,90,91], in northern regions where temperatures are less/not suitable for temperate species [92], and their specific metabolic adaptations can be used to their advantage [21,29]. Moreover, the similar growth rate and Fx content of *F. cylindrus* to that of *P. tricornutum* under similar conditions (f/2, 30 µmol photons m^−2^ s^−1^ PAR, but 19 ± 1 °C, [93]) suggests that the optimisation of the *F. cylindrus* culturing process could significantly increase its Fx productivity, as reported for temperate species [3,4,16,41,42,78,94]. Such approaches are still in their infancy in polar strains [90] and need to be further investigated. Productivity can highly vary depending on growth conditions (Appendix A) [3,13]. The most important growing factors are the culturing (batch, (semi-)continuous) and ‘feeding’ (auto-, mixo- and hetero-trophy) modes, the light (photoperiod, intensity, spectrum) and nutrient availability [3]. Temperature is not as critical for temperate as it is for polar strains [29,46,91]; as demonstrated in our study, up to a certain maximal limit (different from a species to another), some degrees above 0 °C dramatically boost the growth rate and Fx productivity, and may probably allow the cells to use higher light doses for stronger yields (not tested here). In our study, we tested some of these essential growing factors: semi-continuous and batch culturing mode, autotrophy, two temperatures, more or less nutrient-enriched f/2 medium, with a strong focus on light, examining in details changes in photoperiods, light intensity and spectra.

The light dose needs to be adjusted for the best balance between a fast growth rate (supported by a strong light dose up to a certain limit to avoid harmful excess) and a high Fx content (supported by a low light dose), which can be achieved by adjusting the photoperiod and intensity. Due to the high latitude and low-light natural habitat resulting from the snow and ice cover over the water column, most polar strains are adapted to a large range of photoperiods but to a limited range of intensities: this factor thus needs to be carefully adjusted. In our study, a PUR of 11.7 μmol photons m^−2^ s^−1^ with a 12 h light:12 h dark photoperiod appeared to be the sweet spot for *F. cylindrus* growth at 0 °C. Additionally, in the context of indoor growth and/or light supplemented cultures, the use of monochromatic lights solely or mixed proved to be efficient in increasing Fx production [35,41,42,95,96]. Several studies especially reported how the manipulation of the blue:red light ratio can increase Fx production up to 2-fold [35,42,78,95]. Here, we tested blue (445 nm) and red (630 nm) wavelengths alone, and one blue:red ratio (3:1) when using ‘white’ light. With blue light alone, we observed a positive effect on Fx (and Ddx+Dtx) production while reducing light energy consumption by a factor of 2 (see above Section 3.3). Based on these results, as well as on the unique red light response (generating a strong Fx accumulation), future work needs to explore how the manipulation of the blue:red ratio could further improve the Fx production in *F. cylindrus*. Obviously, the sea ice and underneath ice light spectra, which show a very specific blue:red ratio depending on ice type [43], will be a solid starting point.

The unselective enrichment of the culture medium (f > f/2, Figure 8) additionally supported faster growth rate, stronger biomass accumulation and higher Fx production in *F. cylindrus*, independent of the temperature. Medium enrichment is a common way of increasing Fx production [94] (by up to 3-fold [97]), either through nitrogen [32,33,34,76,93,98], iron [97], silicates and/or CO_2_ enrichment of the cultures [95,99]. Additionally, nutrient shortage, and to a larger extent environmental stresses (e.g., high salinity, UV exposure, etc. [100,101,102]), were shown to boost the production of Fx, another path that could be investigated in polar strains.

Finally, and as a complement to the optimization of growing conditions, Fx productivity in *F. cylindrus* could be further improved by means of genetic engineering [17], as demonstrated in *P. tricornutum* [15,18], where the Fx content was increased by 1.5- to 1.8-fold in this way [16]. In parallel, strain selection is equally important [3,4,13,35,36]. We have chosen *F. cylindrus* based on the wealth of genomic, transcriptomic and physiological data already available [47,48,50,51,52,53], but other species with a higher intracellular Fx content [46,62,103] could be used. We also found that *T. gravida* and *C. neogracilis* produce and accumulate high amounts of Fx and Ddx+Dtx (Table 3) [51,53]. These are promising candidates for future investigations, and the approach presented here will serve as a solid basis for the next steps in the improvement of xanthophyll pigment production.

## 4. Materials and Methods

### 4.1. Culturing Conditions

Culturing and all experiments were performed in a climate-controlled ‘cold’ laboratory where temperature was set to match the experimental temperature (i.e., 0 °C or 7 °C) and humidity was permanently controlled to avoid condensation (Dew point below −13 °C). Culturing and experiments were performed in axenic conditions with sterile and/or autoclaved equipment handled under a laminar flow hood inside the ‘cold’ laboratory to avoid contamination. Axenic *Fragilariopsis cylindrus* (Figure 9; CCMP3323, isolated in Antarctica; −64.08° N −48.7033° W) was grown in natural sterile seawater (sampled in Baffin Bay, Canadian Arctic, 67.48 N; 63.79 W) enriched with f/2 or f medium (for detailed composition, see Appendix A and Guillard [104]) depending on the experiment. The sea water was prefiltered through a polypropylene 1 μm filter (Polypropylene felt filter bag 18-1/2L, 1 μm, Cole-Parmer, Montréal, QC, Canada) and a PolyCap 0.2 μm (Whatman™, UK). Cultures were pre-acclimated to each experimental light and temperature conditions for at least 3 weeks before the start of the experiment. They were gently stirred with a magnetic stirrer and aerated with air bubbling filtered through an activated carbon filter and a 0.3 μm HEPA filter (Carbon CAP, HEPA-Vent, Whatman™, Maidstone, UK). Cells were grown in triplicate in 3 L jacketed cylinder reactors that were additionally temperature-controlled (0 °C or 7 °C) thanks to the circulation of thermostated ethylene glycol through the jacket [52]. Light was supplied uniformly by a custom-built illumination system comprising an array of LEDs of different wavelengths (LXML-PR01, 445 nm; LXML-PB01, 470 nm; LXML-PM01, 505 nm; LXML-PM01, 530 nm; LXM2-PD01, 630 nm; LXM3-PD01, 660 nm; LXML-PD01, 4100 K; LXML-PL01, amber; LUXEON REBEL, LUMILEDS, Aachen, Germany) that allowed the modulation of light intensity, light spectrum and photoperiod (Appendix A). Photosynthetically available radiation (PAR) was measured continuously in the centre of each reactor using a 4π PAR sensor (QSL 2101, Biospherical Instruments Inc., San Diego, CA, USA). This continuous PAR monitoring allowed us to automatically maintain irradiance inside the reactor at a targeted value, which would otherwise vary depending upon cell concentration and culture optical density. In our study, 30 µmol photons m^−2^ s^−1^ of PAR was selected based on previous works showing that this intensity is optimal for *F. cylindrus* growth [52]. The light photoperiod, intensity and spectrum were experimentally varied and their effect on xanthophyll synthesis tested (see Section 2).

### 4.2. Experimental Conditions and Sampling Plan

All samplings and measurements were performed in the ‘cold’ laboratory under a green light of low intensity to limit the alteration of cell photosynthetic activity and pigments due to ambient light. We designed two distinct experiments: (1) a first series of experiments to define the best growing conditions for fucoxanthin (Fx) productivity in *F. cylindrus*, by exploring the effects of varying light spectrum, intensity and temperature, and (2) a second series to compare *F. cylindrus* Fx productivity with the one of diatom temperate counterparts.

#### 4.2.1. Optimization of Fucoxanthin (Fx) Productivity

*F. cylindrus* cultures were grown in f/2 medium, at 0 °C and 7 °C under different photoperiods, PAR intensities and spectra (Table 1). Importantly, and contrary to many studies conducted in the field, cultures were kept optically thin, in order to control the light field inside the reactors, by a daily dilution with fresh f/2 medium (semi-continuous cultivation) to maintain a cell concentration of 10^6^ cells mL^−1^. The so-called ‘white’ light spectrum used here was built to be as close as possible to the spectrum observed at the horizon of bottom sea ice (Appendix A). The monochromatic blue and red lights had a full width half maximum of 20 nm centred at 450 and 660 nm, respectively. Each experiments lasted 72 h with samplings twice a day, 1 h after the beginning of the light phase and 1 h before the end, for performing cell counts, and analysing elemental composition, pigment analyses and photosynthetic performance (i.e., rapid light curves, see below). Cultures were diluted after the second sampling point, and 30 min after the dilution (to allow homogenisation and stabilisation of the culture) samples were collected for cell counts and pigment analyses.

#### 4.2.2. Comparison of *F. cylindrus* Fx Productivity with Temperate Counterparts

In order to directly compare the potential of *F. cylindrus* to synthesize Fx with one of the temperate counterparts, we grew it in the same conditions (but temperature and light intensity due to specific polar adaptations) as in Wang et al. [35]. *F. cylindrus* cultures were inoculated with 10^6^ cells mL^−1^ and grown in batch mode in f/2 and f media, at 0 °C and 7 °C, and under a ‘white’ light spectrum of 30 µmol photons m^−2^ s^−1^ PAR. Here, the light intensity was set after the inoculation of the culture and left unchanged throughout the experiment (i.e., without adjustment for optical density increasing). Sampling was performed every two days for cell counting, and the analysis of elemental composition and pigments.

### 4.3. PAR, PUR and Determination of Energy Consumption

Differing from PAR, PUR is the photosynthetically useable radiation (in μmol photons m^−2^ s^−1^), which is the part of PAR that is effectively absorbed by *F. cylindrus* cells and which varies depending upon their pigment composition. For determining *F. cylindrus* PUR, its absorption spectrum was measured using a double beam spectrophotometer (Lambda 850, PerkinElmer, Waltham, MA, USA) equipped with an integrative sphere of 15 cm in diameter (150 mm RSA ASSY, PerkinElmer, Waltham, MA, USA), and following the procedure described by Babin and Stramski [105] (Appendix A). OD (λ) was measured between 380 nm and 760 nm with 1 nm increments in a cuvette (1 × 1 cm) containing a 3 mL *F. cylindrus* culture sample. From this measurement, the in vivo specific absorption spectrum of *F. cylindrus a_ph_(λ)* ((mg chla m^−3^)^−1^) was calculated as: $a_{ph}=\ln\left(OD\left(\lambda \right)\;\right)/\left[Chl\;a\right]\times l$, with *l* = 0.01 m. Finally, the determination of PUR was carried out as follows (Equation (1)) [106]:(1)$$\mathrm{PUR}=\int_{400}^{700}A(\lambda )\frac{E\left(\lambda \right)d\lambda }{d\lambda }d\lambda$$
where *A(λ)* is *a_ph_(λ)* normalized to its maximum (*A(λ)* ≡ 1, *λ* = 440 nm) and *𝐸(λ)* is the emission spectrum of the light source.

To calculate the proper amount of PAR necessary to maintain the same PUR, between light spectra, we used the effective absorption coefficient of *F. cylindrus* ( $a_{eff}={\bar{a}}_{ph}\;/\mathrm{PAR}$, where ${\bar{a}}_{ph}=\int_{400}^{700}a_{ph}(\lambda )\frac{E\left(\lambda \right)d\lambda }{d\lambda }d\lambda$ and $\mathrm{PAR}=\int_{400}^{700}\frac{E\left(\lambda \right)d\lambda }{d\lambda }d\lambda$) for a specific light spectrum and made sure that the specific absorption coefficient (${\bar{a}}_{ph})$ remained the same between treatment with similar PUR by adjusting the PAR by ${\mathrm{PAR}}_{2}=\frac{a_{eff}1}{a_{eff}2}\times \mathrm{PAR}{\;}_{1}$.

In parallel, the energy consumption of the lighting system was measured for the duration of each experiment. The consumption was then divided by the volume of the culture to obtain an energy consumption in Wh day^−1^ L^−1^, for an lit area of 292 cm^−2^ L^−1^.

### 4.4. Cell Concentration and Growth Rate

Cell concentration was measured using a particle sizing and counting analyser (Multisizer 4 Coulter Counter, Beckman Coulter, Brea, CA, USA). Analyses were performed for cells between 2 and 15 μm in size. A dilution of the sample was performed before analysis in order to remain within the cell concentration measurement range of the analyser. Dilution was carried out with the electrolyte of the analyser, i.e., salted Milli-Q™ water (35 g L^−1^. NaCl). The growth rate (μ, in day^−1^) was calculated according to Andersen [107], between dilutions in semi-continuous growing conditions and until the end of the expontential growth phase for the batch growing conditions.

### 4.5. Particulate Organic Carbon and Nitrogen Determination, and Algal Biomass Dry Weight

The total particulate carbon (TPC), total particulate nitrogen (TPN), and the dry algal biomass were determined by filtering culture samples on pre-burned (450 °C for 4 h) 25 mm GF/F glass-fibre filters (Whatman™, Maidstone, UK). To remove the seawater salt from the filters without inducing an osmotic stress for *F. cylindrus* cells, the filters were rinsed with 40 mL of 0.5 M ammonium formate. The pre-combusted filters were weighted before filtration and a second time after the algal biomass was dried out in an oven at 60 °C for 48 h to determine their dry weight (DW). The CHN analyses were performed with a PerkinElmer 2400 Series II CHNS/O elemental analyser (PerkinElmer, Waltham, MA, USA), and acetanilide was used as a standard.

### 4.6. Pigment Extraction and Quantification

Pigments were quantified by high-performance liquid chromatography (HPLC Agilent 1260 Infinity system, Agilent, Santa Clara, CA, USA) equipped with a C8 reverse-phase column (Zorbax Eclipse XDB-C8, Agilent, USA), following the extraction and analytical procedure of Ras et al. [108]. The entire procedure was performed on ice in the ‘cold’ laboratory. Samples of *F. cylindrus* cultures were filtered on 25 mm GF/F glass-fibre filters (Watman™, Littls Chalfont, UK) and immediately frozen in liquid nitrogen and stored at −80 °C until further analysis. Samples were extracted in 3 mL of extraction buffer (100% methanol HPLC grade and vitamin E acetate (15.9 μM, Sigma) as internal standard) stored at −20 °C by grounding the filters by means of sonication (15 s followed by a 1 h storage at −20 °C). After centrifugation (15 min, 4500 rpm, 0 °C), the supernatants were filtered through 25 mm GA55 glass-fibre filters (Cole-Parmer, Canada). Then, 700 μL of the filtered supernatants was transferred to chromatography vials complemented with 300 μL of a tetrabutylammonium acetate (TBAA) solution (28 mM) for polarization. A final volume of 50 μL was injected into the HPLC system. Separation was achieved with a gradient between a solution (A) of TBAA 28 mM:methanol 100% (30:70, v:v) and a solution (B) of 100% methanol according to Ras et al. [108]. Fucoxanthin, diadinoxanthin and diatoxanthin were detected at 450 nm using a diode array detector (see Appendix A for a typical chromatogram), while vitamin E acetate was detected at 220 nm and chlorophyll *a* (Chl *a*) at 640 nm. Pigment quantification was carried out using pigment standards provided by D.H.I. Water & Environment (Horsholm, Denmark) and the internal standard (vitamin E acetate).

### 4.7. Photosynthetic Performances

A blue light source (λ = 460 ± 20 nm) pulse amplitude modulated (PAM) fluorometer (WATER-ED/B, Walz, Germany) was used to measure the photosynthetic performance of *F. cylindrus* cells on dark-acclimated (30 min) samples. Rapid light curves (RLCs) were performed with 12 steps of 30 s and of increasing light intensity from 0 to 611 µmol photons m^−2^ s^−1^. The dark-acclimated photochemical efficiency of photosystem II-PS II (F_V_/F_M_) was calculated as in Equation (2):(2)$$F_{V}/F_{M}=\frac{F_{M}-F_{0}}{F_{M}}$$
where F_0_ and F_M_ are, respectively, the minimum and maximum levels of dark-acclimated chlorophyll fluorescence. The relative electron transport rate (rETR) was calculated as in Equation (3):(3)$$\mathrm{rETR}=E\;\times \frac{F_{M}^{\prime }-\;F\prime }{F_{M}^{\prime }}$$
where E is the actinic light intensity, and F′ and F_M_′ are, respectively, the actual and maximum levels of light acclimated chlorophyll fluorescence. The determination of rETR for each of the 12 intensities of the RLCs allowed us to build rETR vs. E curves that were fitted according to Eilers and Peeters [109] in order to extract photosynthetic parameters (see Barnett et al. [82]): rETR_max_ (the maximum relative electron transport rate), α (the maximum light efficiency use), E_k_ (rETR_max_/α, the light saturation coefficient or ‘photoacclimation’ parameter) and E_opt_ (the optimal light intensity for reaching rETR_max_).

The output of the RLCs also allowed us to calculate the non-photochemical quenching (NPQ) as in Equation (4):(4)$$\mathrm{NPQ}=\frac{\left(F_{M}-F_{M}^{\prime }\right)}{F_{M}^{\prime }}$$

The maximal NPQ (NPQ_max_) was the maximal value obtained for the highest RLC step (611 µmol photons m^−2^ s^−1^), and the non-photochemical quenching at growing PUR intensity (NPQ_gE_) was the NPQ obtained for the RLC step with the closest PUR intensity to the growing PUR intensity available.

The partitioning of absorbed excitation energy in photosystem II (PSII) was determined by the complementary PSII quantum yields method [110,111], where Y_PSII_ ($\frac{F_{\mathrm{MgE}}-F_{\mathrm{gE}}}{F_{\mathrm{MgE}}}$) represents the fraction of energy photochemically converted through PSII, Y_NPQ_ ($\frac{F_{\mathrm{gE}}}{F_{\mathrm{MgE}}}\frac{F_{\mathrm{gE}}}{F_{M}})$ represents the fraction of energy dissipated in form of heat via the regulated NPQ, and Y_NO_ ($\frac{F_{\mathrm{gE}}}{F_{M}}$) represents the fraction of energy that is passively dissipated in form of heat and fluorescence [110,111,112]. F′ and F_M_′ were measured during RLC at the light step with the closest PUR intensity to the growing PUR intensity.

### 4.8. Statistical Analysis

A 1-way ANOVA followed by Tukey’s HSD post hoc test was used to test differences in the measured parameters’ means between treatments (see Figure 2, Figure 3, Figure 4, Figure 5, Figure 6, Figure 7, Figure 8 and Appendix A). A 3-way ANOVA followed by Tukey’s HSD post hoc test was used with the growth light spectrum, intensity and temperature as independent factors to determine the effect of these factors and their interaction over *F. cylindrus* physiology and productivity (Appendix A).

## 5. Conclusions

The present work supports the possibility of using polar diatoms as an efficient cold and low light-adapted bioresource for xanthophyll pigments, especially usable in Nordic countries to develop new economics based on marine resources [21,23]. Beyond indoor growth, which would require further investigations to reduce energetic costs related to temperature regulation, the present work also highlights the possibility of growing polar diatoms outdoors during winter at temperate latitudes or part of the year (depending on the latitude) in northern areas. Polar species other than *Fragilariopsis cylindrus* that show even stronger potential in fucoxanthin production, such as *Chaetoceros neogracile*, could also be used. We have furthermore demonstrated that, beyond fucoxanthin, polar diatom biomass valorisation can include other bioactive xanthophyll pigments of interest, namely diadinoxanthin+diatoxanthin [6]. Since diatoms are enriched in polyunsaturated fatty acid (PUFAs) of interest, such as eicosapentaenoic acid [4,90,91,96], and phenolic compounds, vitamins and other bioactive metabolites [113,114], polar diatom biomass valorisation beyond pigments could also be considered [115].

## Figures and Tables

**Figure 1 marinedrugs-20-00491-f001:**
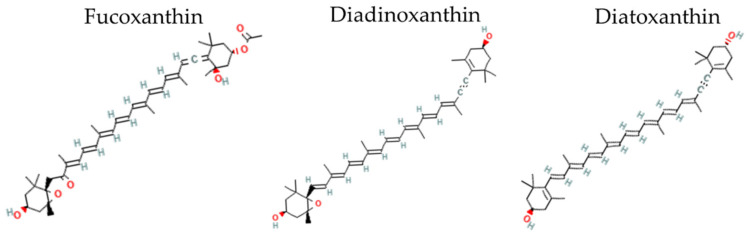
Two-dimensional structure of fucoxanthin, diadinoxanthin and diatoxanthin. Two-dimensional structural images of CID 5281239 (fucoxanthin), 6449888 (diadinoxanthin), 6440986 (Diatoxathin) were obtained from PubChem (https://pubchem.ncbi.nlm.nih.gov, accessed on 27 June 2022).

**Figure 2 marinedrugs-20-00491-f002:**
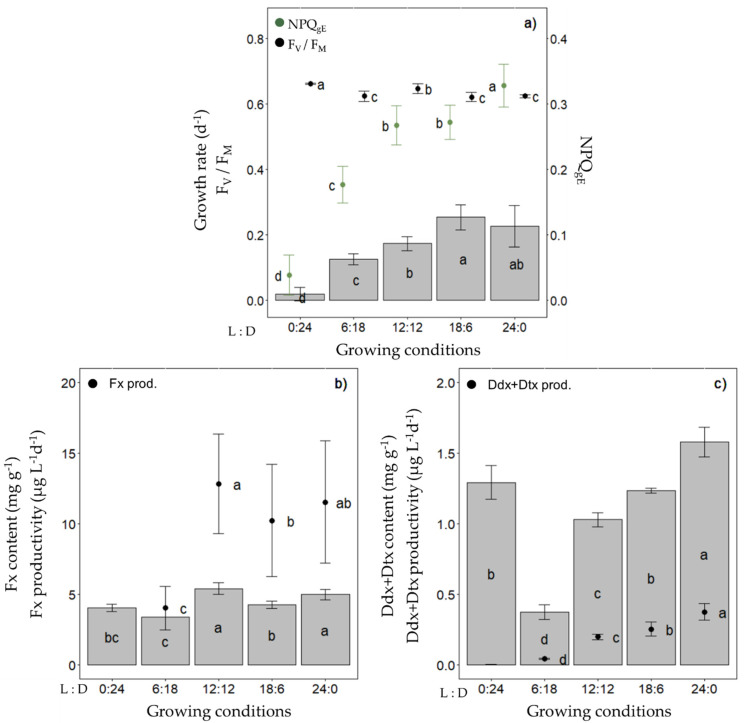
Photosynthetic potential, fucoxanthin (Fx), diadinoxanthin and diatoxanthin (Ddx+Dtx) synthesis in *Fragilariopsis cylindrus* grown under different photoperiods (0 h light:24 h darkness; 6 h L:18 h D; 12 h L:12 h D; 18 h L:6 h D; 24 h L:0 h D) under a ‘white’ spectrum (see Appendix A) with the same photosynthetically usable radiation (PUR) (11.7 µmol photons m^−2^ s^−1^): (**a**) growth rate (bars), dark-acclimated photochemical efficiency (F_v_/F_M_, black dots), effective non-photochemical quenching (NPQ_gE_, green dots); (**b**) Fx content (bars), Fx productivity (dots); (**c**) Ddx+Dtx content (bars), Ddx+Dtx productivity (dots). Data are the mean values *n* = 3 ± SD. Letters represent clusters of non-significantly different means for the corresponding parameter, with the letter ‘a’ representing the highest mean values and the other letters following in alphabetic order.

**Figure 3 marinedrugs-20-00491-f003:**
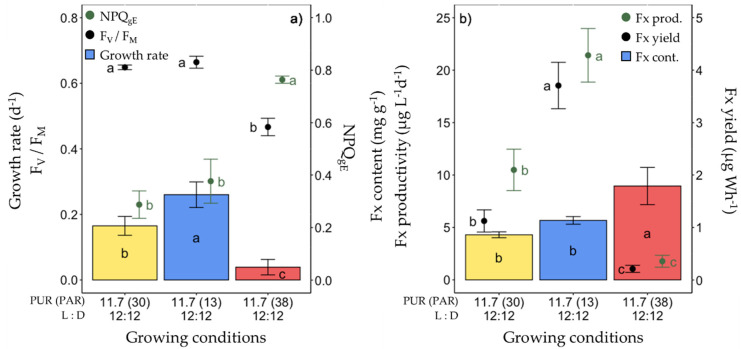
Growth, photosynthetic potential and fucoxanthin (Fx) synthesis in *Fragilariopsis cylindrus* grown under different light spectra (‘white’(yellow), blue (445 nm, blue) and red (660 nm, red)) with the same photosynthetically usable radiation (PUR) (11.7 µmol photons m^−2^ s^−1^) and photoperiod of 12 h L:12 h D: (**a**) growth rate (bars), maximum photochemical efficiency (F_v_/F_M_, black dots), non-photochemical quenching at growing PUR intensity (NPQ_gE_, green dots), (**b**) Fx content (bars), Fx productivity (green dots) and Fx yield (black dots). Data are the mean values *n* = 3 ± SD; see Appendix A for all values and see Table 2 for parameter definitions. Letters represent clusters of non-significantly different means for the corresponding parameter, with the letter ‘a’ representing the highest mean values and the other letters following in alphabetic order.

**Figure 4 marinedrugs-20-00491-f004:**
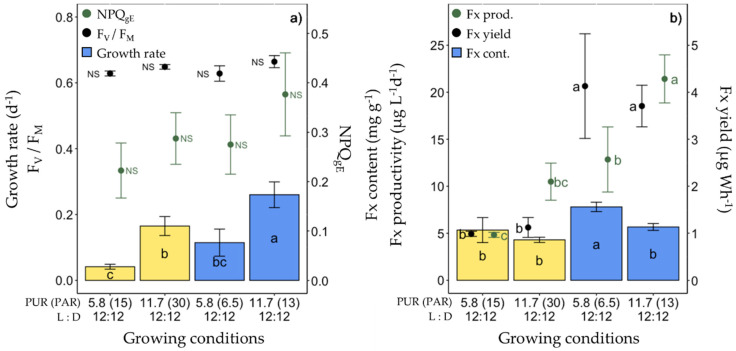
Growth, photosynthetic potential and fucoxanthin (Fx) synthesis in *Fragilariopsis cylindrus* grown under two light spectra (‘white’ (yellow bars), blue (445 nm, blue bars)) and two photosynthetically usable radiation (PUR) levels (5.8 and 11.7 µmol photons m^−2^ s^−1^) with the same photoperiod of 12 h L:12 h D: (**a**) growth rate (bars), maximum photochemical efficiency (F_v_/F_M_, black dots), non-photochemical quenching at growing PUR intensity (NPQ_gE_, green dots), (**b**) Fx content (bars), Fx productivity (green dots) and Fx yield (black dots). Data are the mean values *n* = 3 ± SD; see Appendix A for all values and see Table 2 for parameter definitions. Letters represent clusters of non-significantly different means for the corresponding parameter, with the letter ‘a’ representing the highest mean values and the other letters following in alphabetic order. NS represent non statically different means for the parameter across the treatments.

**Figure 5 marinedrugs-20-00491-f005:**
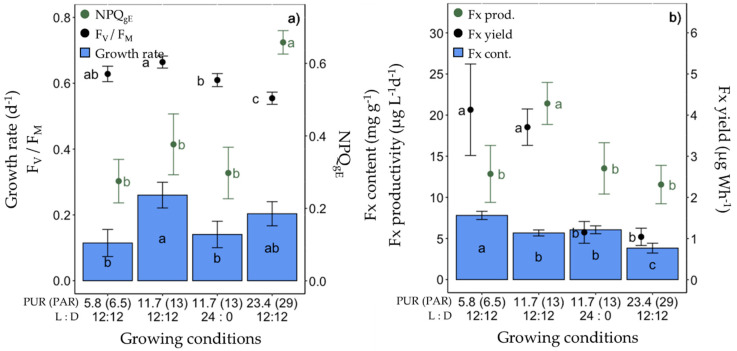
Growth, photosynthetic potential and fucoxanthin (Fx) synthesis in *Fragilariopsis cylindrus* grown under different photosynthetically usable radiation (PUR) levels (5.8, 11.7 and 23.4 µmol photons m^−2^ s^−1^) and photoperiods (12 h L:12 h D and continuous light, 24 h L:0 h D) with the same light spectra (445 nm, blue): (**a**) growth rate (bars), maximum photochemical efficiency (F_v_/F_M_, black dots), non-photochemical quenching at growing PUR intensity (NPQ_gE_, green dots), (**b**) Fx content (bars), Fx productivity (green dots) and Fx yield (black dots). Data are the mean values *n* = 3 ± SD; see Appendix A for all values and see Table 2 for parameter definitions. Letters represent clusters of non-significantly different means for the corresponding parameter, with the letter ‘a’ representing the highest mean values and the other letters following in alphabetic order.

**Figure 6 marinedrugs-20-00491-f006:**
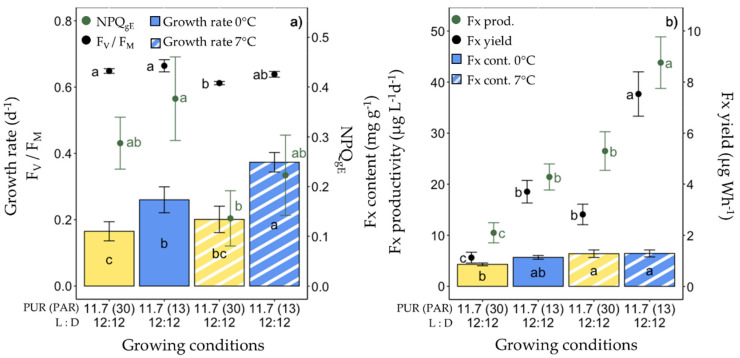
Growth, photosynthetic potential and fucoxanthin (Fx) synthesis in *Fragilariopsis cylindrus* grown under two light spectra (‘white’ (yellow bars), blue (445 nm, blue bars)) and two temperatures (0 °C (plain bars), and 7 °C (striped bars)) with the same photosynthetically usable radiation (PUR) levels (11.7 µmol photons m^−2^ s^−1^) and photoperiod of 12 h L:12 h D: (**a**) growth rate (bars), maximum photochemical efficiency (F_v_/F_M_, black dots), non-photochemical quenching at growing PUR intensity (NPQ_gE_, green dots), (**b**) Fx content (bars), Fx productivity (green dots) and Fx yield (black dots). Data are the mean values *n* = 3 ± SD; see Appendix A for all values and see Table 2 for parameter definitions. Letters represent clusters of non-significantly different means for the corresponding parameter, the letter ‘a’ being the highest mean values and the other letters following in alphabetic order.

**Figure 7 marinedrugs-20-00491-f007:**
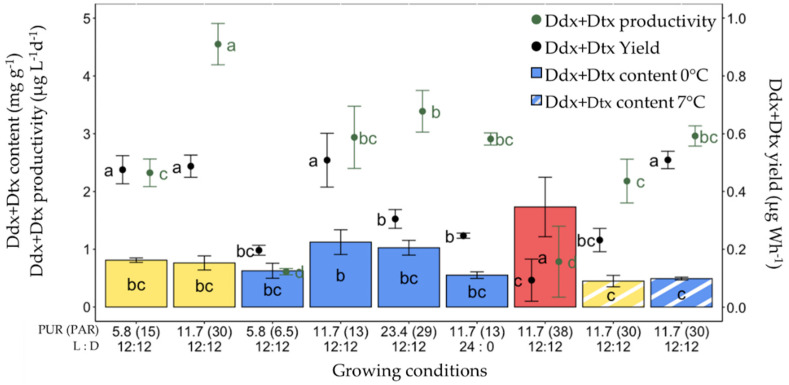
Diadinoxanthin and diatoxanthin (Ddx+Dtx) synthesis *in Fragilariopsis cylindrus* grown under different light spectra (‘white’ (yellow bars), blue (445 nm, blue bars) and red (660 nm, red bars), photosynthetically usable radiation (PUR) levels (5.8, 11.7, 23.4 µmol photons m^−2^ s^−1^), photoperiods (12 h L:12 h D and continuous light, 24 h L:0 h D), and temperatures (0 °C (plain bars), and 7 °C (striped bars)). Data are the mean values *n* = 3 ± SD; see Appendix A for all values and see Table 2 for parameter definitions. Letters represent clusters of non-significantly different means for the corresponding parameter, with the letter ‘a’ representing the highest mean values and the other letters following in alphabetic order.

**Figure 8 marinedrugs-20-00491-f008:**
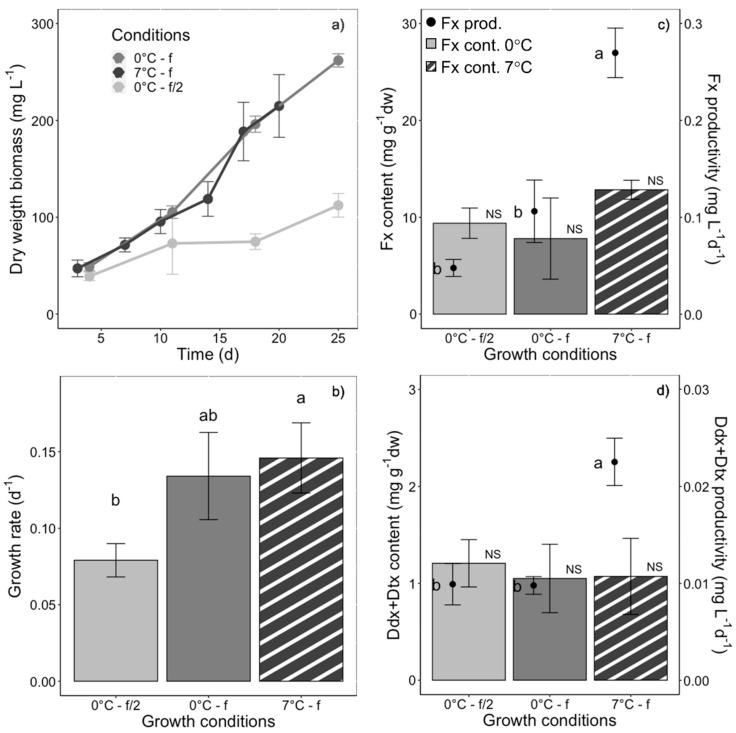
Growth, fucoxanthin (Fx), diadinoxanthin and diatoxanthin (Ddx+Dtx) synthesis in *Fragilariopsis cylindrus* grown in batch culturing mode at 0 °C (plain bars) and 7 °C (striped bars), in f/2 (light grey) and f medium (medium and dark grey), and under the same photosynthetically usable radiation (PUR) (11.7 µmol photons m^−2^ s^−1^) and photoperiod of 12 h light:12 h darkness: (**a**) biomass dry weight increase over days, (**b**) growth rate, (**c**) Fx content (bars), Fx productivity (dots), (**d**) Ddx+Dtx content (bars), Ddx+Dtx productivity (dots). f/2 and f refer to the culturing medium enriched at half and full concentration. Data are the mean values *n* = 3 ± SD; see Table 2 for parameter definitions. Letters represent clusters of non-significantly different means for the corresponding parameter, with letter ‘a’ representing the highest mean values and the other letters following in alphabetic order. NS represents non statistically different means for the parameter across the treatments.

**Figure 9 marinedrugs-20-00491-f009:**
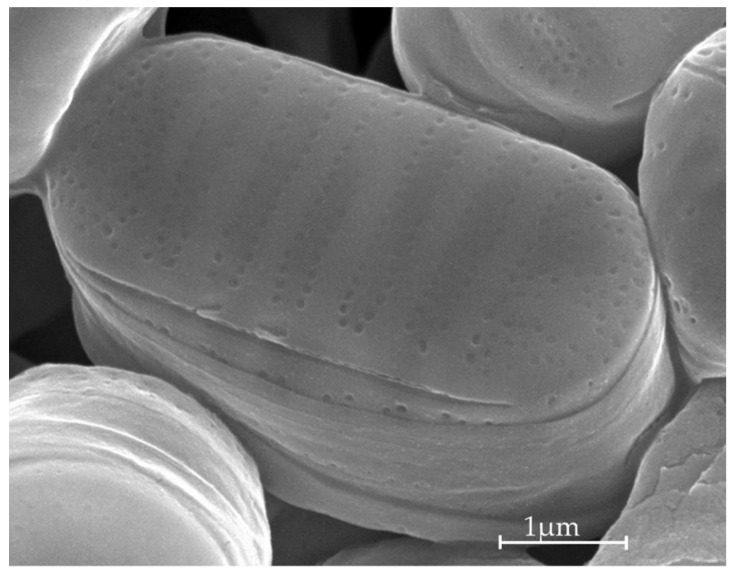
*Fragilariopsis cylindrus* (CCMP 3323) scanning electron microscope picture. Taken by Adèle Luthi-Marie and Suzie Côté at the microanalysis laboratory of University Laval.

**Table 1 marinedrugs-20-00491-t001:** The temperature and light conditions (photoperiod, spectrum and intensity) investigated in this study, and the corresponding daily energy consumption for lighting the 2.7 L reactors.

Temperature (°C)	Photoperiod (Light:Dark)	Spectrum	PUR (µmol Photons m^−2^ s ^−1^)	PAR (µmol Photons m^−2^ s ^−1^)	Daily Energy Consumption (for 2.7 L Reactors)
0	12 L:12 D	‘White’	11.7	30	25.2 W
0	12:12	‘White’	5.8	15	13.2 W
7	12:12	‘White’	11.7	30	25.4 W
0	12 L:12 D	Blue (445 nm)	11.7	13	15.6 W
0	12:12	Blue	5.8	6.5	8.4 W
0	24 L:00 D	Blue	11.7	13	31.8 W
0	12:12	Blue	23.4	29	30 W
7	12:12	Blue	11.7	13	15.7 W
0	12 L:12 D	Red (660 nm)	11.7	38	22.8 W

**Table 2 marinedrugs-20-00491-t002:** Synthesis of all parameters measured in this study, their units, definition, meaning and measurement method.

Parameter	Unit	Definition	Meaning	Measurement
F_0_	No units	Minimum PSII Chl fluorescence yield	Used to calculate Fv/Fm	Rapid Light Curves-RLCs, after 30 min of dark acclimation
F_M_	No units	Maximum PSII Chl fluorescence yield	Used to calculate Fv/Fm, NPQ, Y_NPQ_, Y_NO_	RLCs, during a saturating pulse after 30 min of dark acclimation
F′	No units	F for illuminated cells	Used to compute rETR	RLCs, after 30 s of illumination at specific light intensity-E
F_M_′	No units	F_M_ for illuminated cells	Used to compute NPQ and rETR	RLCs, during a saturating pulse after 30 s of illumination at specific E
F_gE_	No units	F for cells illuminated with the growing light gE	Used to calculate Y_PSII_, Y_NPQ_ and Y_NO_	RLCs, after 30 s of illumination at E the closest to the growing light gE
F_MgE_	No units	F_M_ for cells illuminated with growing light gE	Used to compute Y_PSII_ and Y_NPQ._	RLCs, during a saturating pulse after 30 s of illumination at E the closest to the growing light gE
F_V_/F_M_	No units	Maximum photosynthetic efficiency of PSII; F_V_ = F_M_ – F_0_	The dark-acclimated photochemical efficiency of photosystem II	/
rETR	μmol electrons m^−2^ s^−1^	Relative photosynthetic electron transport rate $=E\;\times \frac{F_{M}^{\prime }-\;F\prime }{F_{M}^{\prime }}$	Effective quantum yield of photochemistry vs. E	RLCs
NPQ	rel. unit.	Non-photochemical quenching $=\frac{\left(F_{M}-F_{M}^{\prime }\right)}{F_{M}^{\prime }}$	Estimates the photoprotective dissipation of excess light energy	RLCs
rETR_max_	μmol electrons m^−2^ s^−1^	rETR-E curve asymptote	Maximum relative photosynthetic electron transport rate	Derived from fitted rETR-E curves measured with RLCs
NPQ_max_	rel. unit.	NPQ-E curve asymptote	Maximum non-photochemical quenching	RLCs
NPQ_gE_	rel. unit.	Non-photochemical quenching $\frac{\left(F_{M}-F_{\mathrm{MgE}}\right)}{F_{M}\mathrm{gE}}$	Estimates of the photoprotective dissipation of excess energy under the growing light intensity gE	RLCs
Y_PSII_	rel. unit.	Quantum yield of photochemical energy conversion in PSII = ($\frac{F_{\mathrm{MgE}}-F_{\mathrm{gE}}}{F_{\mathrm{MgE}}}$)	Estimates the fraction of energy photochemically converted through PSII	RLCs
Y_NPQ_	rel. unit.	Quantum yield of regulated non-photochemical energy loss in PSII = ($\frac{F_{\mathrm{gE}}}{F_{\mathrm{MgE}}}-\frac{F_{\mathrm{gE}}}{F_{M}}$)	Estimates the fraction of energy dissipated as heat via the regulated NPQ	RLCs
Y_NO_	rel. unit.	Quantum yield of non-regulated non-photochemical energy loss in PSII = ($\frac{F_{\mathrm{gE}}}{F_{M}}$)	Estimates the fraction of energy that is passively dissipated as heat and fluorescence	RLCs
**Parameter**	**Unit**	**Definition**	**Meaning**	**Method**
Chl *a*	mg L^−1^	Volumetric chlorophyll *a* concentration	Chl *a* concentration	HPLC pigments quantification
Fx	mol 100 mol^−1^	Fucoxanthin	Fx for 100 mol of Chl *a*	HPLC pigments quantification
Ddx	mol 100 mol^−1^	Diadinoxanthin	Ddx for 100 mol of Chl *a*	HPLC pigments quantification
Dtx	mol 100 mol^−1^	Diatoxanthin	Dtx for 100 mol of Chl *a*	HPLC pigments quantification
Ddx+Dtx	mol 100 mol^−1^	Xanthophyll pool	Ddx+Dtx for 100 mol of Chl *a*	HPLC pigments quantification
Cells	cells mL^−1^	Algae cellular density	Count of cells per volume of culture	Particle sizer and counter
μ	d^−1^	Growth rate =ln(nn+1)/Δt	Population division rate per day	Calculated every 24 h
P	Wh	Power consumption	Power consumption of the lightning source for a culture of 2.7 L.	Consumption measured at the outlet for a 24 h period
C	mg L^1^	Total particulate carbon	Carbon content of the particulate fraction of the culture	CHN analyser
N	mg L^−1^	Total particulate nitrogen content	Nitrogen content of the particulate fraction of the culture	CHN analyser
DW	mg L^−1^	Dry weight	Dry weight of the particulate fraction of the culture	Gravimetry
C/N	g g^−1^	Carbon:nitrogen ratio	/	/
Cellular C	pg cell^−1^	Intracellular carbon content	/	/
Cellular N	pg cell^−1^	Intracellular nitrogen content	/	/
Cellular Chl *a*	pg cell^−1^	Intracellular chlorophyll *a* content	/	/
Fx cont.	mg g^−1^	Fucoxanthin content	Fucoxanthin content per dry weight of algae cells	/
Ddx+Dtx cont.	mg g^−1^	Diadinoxanthin+diatoxanthin content	Diadinoxanthin+diatoxanthin content per unit of dry weight of algae cells	/
Fx prod.	µg L^−1^ day^−1^	Fucoxanthin productivity	Fucoxanthin produced per day in culturing conditions	/
Ddx+Dtx prod.	µg L^−1^ day^−1^	Diadinoxanthin+diatoxanthin productivity	Diadinoxanthin+diatoxanthin produced per day in culturing conditions	/
Fx yield	µg Wh	Fucoxanthin production	Fucoxanthin produced per unit of energy consumed	/
Ddx+tx yield	µg Wh	Diadinoxanthin+diatoxanthin production	Diadinoxanthin+diatoxanthin produced per unit of energy consumed	/

**Table 3 marinedrugs-20-00491-t003:** Growth rate, fucoxanthin, and diadinoxanthin and diatoxanthin (Ddx+Dtx) productivity of exponentially growing cells acclimated to 15 and 50 µmol photons m^−2^ s^−1^ PAR (photosynthetically available radiation) in several polar diatom strains. Data are the mean values *n* = 3 ± SD.

	PAR (µmol Photons m^−2^ s^−1^)	Growth Rate (Day^−1^)	Fucoxanthin Productivity (µg L^−1^ Day^−1^)	Ddx+Dtx Productivity (µg L^−1^ Day^−1^)
*Nitzschia* *frigida*	15	0.17 ± 0.01	32.4 ± 0.9	2.82 ± 0.08
	50	0.12 ± 0.01	10.5 ± 0.9	1.72 ± 0.15
*Fragilariopsis* *cylindrus*	15	0.1 ± 0.03	4.89 ± 2.21	2.34 ± 0.78
	50	0.25 ± 0.05	11.93 ± 3.39	5.45 ± 1.55
*Thalassiosira* *gravida*	10	0.21 ± 0.03	32.5 ± 5.5	1.12 ± 0.19
	50	0.32 ± 0.01	112.6 ± 15.5	8.86 ± 1.22
*Chaetoceros* *neogracilis*	15	0.55 ± 0.01	200.4 ± 8.6	4.65 ± 0.13
	50	0.62 ± 0.03	261.7 ± 15.7	32.38 ± 1.39
*Chaetoceros* *gelidus*	15	0.20 ± 0.07	35.4 ± 9.3	2.94 ± 0.77
	50	0.33 ± 0.03	62.7 ± 4.5	13.24 ± 0.94

## Data Availability

The data presented in this study are contained within the article and Appendix A and available on request from the corresponding author.

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
