# Peer review of "Potential for the Production of Carotenoids of Interest in the Polar Diatom Fragilariopsis cylindrus"

_marinedrugs, 2022, doi:10.3390/md20080491_

Round 1

Reviewer 1 Report

In the present study, potentiality to produce target carotenoids (Fx,, Dx,and Dtx)by a polar diatom  Fragilariopsis cylindrus was investigated by manipulating the growth light and temperature. Its oudoor potentiality was also mentioned.

The results were promising since they could reach similar productivity with other diatom species like Phaeodactylum tricornutum under light intensity and temperature.

In the abstract section, no results were provided regarding biomass productivity and xanthophyll content. It would be better to give numbers to pay attention to this study.

The main distraction from this study was the way the results were presented. There are too many supplementary tables which make the reader confused. The Table and Figure captions are too long to be understood. In addition, many of the phrases in the Tables in the Supplementary section are alike, and it is not easy to follow the results in this way. I can understand that there is massive data to be presented but I can recommend the authors to use an Appendix at the end of the Manuscript and avoid repeating the same words etc.

In table 3, it will be better to add the values of P. tricornutum since it is also a promising commercial and high Fx producer species.

The results section is extremely complicated and should be simplified.

On the other hand, the results are well discussed based on the literature.

In addition, I would like to see the chromatogram to see the separation of carotenoids by the C8 column. It should be better to add this chromatogram at least in the Supplementary file. Furthermore, it is not appropriate to use vitamin E acetate as an internal standard. Its absorption profile and behaviors are not the same and it can only be the last chance. The internal standard must be a compound whose characteristics are similar to carotenoids. Any carotenoid that is not present in the diatom species could be a better choice of internal standard. Why did the authors choose it as an internal standard?

Reviewer 2 Report

The study entitled “Potential for the production of carotenoids of interest in the polar diatom Fragilariopsis cylindrus” investigated the ability of a polar microalgal strain to produce fucoxanthin and diadinoxanthin-diatoxanthin under different conditions of daylength, light intensity and spectrum. The results are important, and the data are well represented, however the MS needs critical revision.

  1. The abstract is very general, the main results should be highlighted.
  2. The scientific name of “Fragilariopsis cylindrus” should be italicized in the whole MS.
  3. The last sentence in the introduction is more suited to conclusion.
  4. Results:

-       In all the figures, different type of letters, symbols or numbers for significance should be used. i.e use capital letters for one parameter, and small letters for the other.

-       The significant letters could be included in Table 2.

-       There are no need to represent the significant results in the supplementary tables since they are present in figures.

-       L104, L121, 145: correct units.

-       You should provide the real values (as mean + SD) within text when referring to the results (i.e L 104, L118, etc.).

Please revise the sentence (L103- L106), the Fx content is significantly lower at 18:6 L:D.

  • 5. Discussion:
  •        L 360 “The unique response of F. cylindrus to red light.” Please revise.
  • 6. Materials and methods:

-       The source of the microalgal isolate should be provided

-       The composition of f/2 and f medium should be included in detail along with relevant citations.

-       L 579 what is the humidity values?

-       L 619 does the semi-continuous cultivation was performed by diluting the culture with fresh medium or distilled water?

-       L 635 correct f?.

-       L 650 in vivo- italic

-       L651 correct ( (mg chla m-3)-1)

-       The symbols within the text should be italicized as per the equations.

-       L679 correct choc.

-       L680 Correct “Ammonium Formate” start with small letters.

-       Provide the method for Chl a quantification.

Reviewer 3 Report

Production of carotenoids fucoxanthin and Ddx/Dtx by a polar diatom F. cylindrus under conditions of low temperature and low light intensity has been studied and compared with that of P. tricornutum and other diatom species under temperate conditions. Optimum growing  conditions (light intensity, spectrum and photoperiod length) were investigated.

Authors vary the growing conditions - the photoperiod length, light intensity (PAR, PUR), light spectra (white, blue, red) at two temperatures (0 and 7°C) and using the f/2 and f medium, and measure the carotenoid production, photochemical efficiency, non-photochemical quenching and growth rate. Results are arranged in graphical form supplied by tables and properly discussed.

In my opinion, the only shortcoming is the presentation of results, namely the values of variables other than those represented by bars. Let's take the Fig. 1 a) as an example: the growth rate is represented by bars (left Y axis) in dependence on the L:D ratio on the X axis. One would expect the dots depicting NPQ and Fv/Fm at individual L:D on the verticals going through the bars with the letters a, b, c... and marked by the same letters. Instead, the Table at S1 and the Figure legend says "a being the highest ... and the other letters following in alphabetic order" which I find confusing as to what is meant and most notably why this approach is used. The "b" for NPQ is shown twice in the graphics at 12:12 and 18:6 but only once in S1, etc. I can't distinguish if this is an intent or an error. This approach is used in almost all Figures. In my opinion, a less sophisticated approach to the presentation of results using simple Y-X correspondences would be an improvement, even if there would have to be more graphs (which would probably not be the case). As it is, the reader has to tediously decipher what is really meant, using both the graphics and the supplement table, and it mars the overall positive impression of the manuscript.

There are also some typos, e.g., in L532 (adsjuted).

Reviewer 4 Report

The study has optimized extensively on the production of fucoxanthin and diadinoxanthin-diatoxanthin and the cell growth. The data are inspiring in using the polar strains in bioproduction in the future. The manuscript can be improved by restructuring. The reviewer had difficulty in understanding the meaning of the non-photochemical quenching (NPQ) and dark-acclimated photochemical efficiency (FV/FM). Both were introduced in the Discussion, Line 300-303; Line 310-312. Such information should be provided in the Introduction or Results section. Another minor comment is that it would be better to provide the chemical structures for the two major products fucoxanthin and diadinoxanthin-diatoxanthin and real images of the cells in the figures, which are both informative and interesting to readers.
